# Penalised regression improves imputation of cell-type specific expression using RNA-seq data from mixed cell populations compared to domain-specific methods

Wei-Yu Lin [1], Melissa Kartawinata[2,3], Bethany R. Jebson[2,3], Restuadi Restuadi[2,3], Hannah Peckham[3,4], Anna Radziszewska[3,4], Claire T. Deakin[2,3,5], Coziana Ciurtin[3,4], CLUSTER Consortium[¶], Lucy R. Wedderburn [2,3,5], Chris Wallace [1,6,*]

**1** MRC Biostatistics Unit, Cambridge Biomedical Campus, Cambridge, United Kingdom, **2** Infection, Immunity and Inflammation Research and Teaching Department, UCL Great Ormond Street Institute of Child Health, University College London (UCL), London, United Kingdom, **3** Centre for Adolescent Rheumatology Versus Arthritis at University College London (UCL), University College London Hospital (UCLH) and Great Ormond Street Hospital (GOSH), London, United Kingdom, **4** Division of Medicine, Department of Ageing, Rheumatology & Regenerative Medicine, UCL, London, United Kingdom, **5** National Institute for Health Research (NIHR) GOSH Biomedical Research Centre, London, United Kingdom, **6** Cambridge Institute of Therapeutic Immunology and Infectious Disease (CITIID), Jeffrey Cheah Biomedical Centre, Cambridge Biomedical Campus, University of Cambridge, Cambridge, United Kingdom

¶ Membership of [CLUSTER Consortium] is provided in the Acknowledgements.
* cew54@cam.ac.uk

## Abstract

Gene expression studies often use bulk RNA sequencing of mixed cell populations because single cell or sorted cell sequencing may be prohibitively expensive. However, mixed cell studies may miss expression patterns that are restricted to specific cell populations. Computational deconvolution can be used to estimate cell fractions from bulk expression data and infer average cell-type expression in a set of samples (e.g., cases or controls), but imputing sample-level cell-type expression is required for more detailed analyses, such as relating expression to quantitative traits, and is less commonly addressed. Here, we assessed the accuracy of imputing sample-level cell-type expression using a real dataset where mixed peripheral blood mononuclear cells (PBMC) and sorted (CD4, CD8, CD14, CD19) RNA sequencing data were generated from the same subjects (N=158), and pseudobulk datasets synthesised from eQTLgen single cell RNA-seq data. We compared three domain-specific methods, CIBERSORTx, bMIND and debCAM/swCAM, and two cross-domain machine learning methods, multiple response LASSO and ridge, that had not been used for this task before. We also assessed the methods according to their ability to recover differential gene expression (DGE) results. LASSO/ridge showed higher sensitivity but lower specificity for recovering DGE signals seen in observed data compared to deconvolution methods, although LASSO/ridge had higher area under curves than deconvolution methods. Machine learning methods have the potential to outperform domain-specific methods when suitable training data are available.

**Data availability statement:** The Metadata and processed data that support the findings of this study are available in Zenodo (https://doi.org/10.5281/zenodo.10000430). Source data are provided with this paper.

**Funding:** This research was funded by the MRC (MR/R013926/1 to CW & LRW, paying the salary of W-YL, MK; MC_UU_00002/4, salary to CW), Wellcome Trust (WT220788, salary to CW), the NIHR Cambridge Biomedical Research Centre (BRC-1215-20014 to University of Cambridge, salary to W-YL) Versus Arthritis (Grant: 22084 to LRW), Great Ormond Street Hospital Children's Charity (VS0518 to LRW), and Olivia's Vision (to LRW). LRW, CC, CD, HP and AR were supported by Versus Arthritis awards (grants: 20164 and 21593, paying the salary of CD, HP, AR) to the Centre for Adolescent Rheumatology Versus Arthritis at UCL, UCLH and GOSH. HP was supported by PhD studentship included in a Versus Arthritis award to CC (22203). CD had additional support from Cure JM (GOSH102019). The views expressed are those of the author(s) and not necessarily those of the NHS, the NIHR or the Department of Health. The funders had no role in study design, data collection and analysis, decision to publish, or preparation of the manuscript.

**Competing interests:** I have read the journal's policy and the authors of this manuscript have the following competing interests: The CLUSTER consortium has been provided with generous grants from AbbVie and Sobi. CW receives funding from MSD and GSK and is a part-time employee of GSK. These companies had no involvement in the work presented here.

## Author summary

Numerous studies have demonstrated that gene expression in particular subsets of immune cells plays a critical role in the development of diseases and response to treatment. By profiling gene expression from these cells, we can identify disease-relevant genes, comprehend their functions in the disease or response to treatment, and potentially pave the way for screening and patient stratification for prevention and treatment. However, the current cost of single-cell RNA sequencing is too high for large-scale expression profiling analysis. Therefore, an alternative approach is to computationally estimate cell-type specific expression from mixed cell populations, which has been less explored in the field. With this in mind, we proposed using machine learning approaches, multiple response LASSO and ridge, and applied them to synthesised datasets and real-world data where gene expression was measured in mixed and pure cell populations of the same subjects. We compared them to standard methods in the field, and evaluated the accuracy of predicted expression as well as the ability to reconstruct differentially expressed gene signals. Our results revealed that the LASSO/ridge algorithms performed better than existing methods in recovering differentially expressed gene signals, highlighting their potential applications to impute the cell-type expression.

## Introduction

Tissues are a heterogeneous environment, comprised of various different cell populations. In immune-mediated diseases, gene expression profiling of immune cells has identified subsets of genes characterising disease prognosis [1,2]. This approach enables better discrimination of disease pathogenesis than at mixed cell level [3] motivating study of immune cell transcriptomes in these diseases. Studying cell-type-specific expression has revealed gene expression signatures, e.g., CD8 T cell exhaustion, that predict disease course [4]. However, flow sorting of target cells followed by RNA extraction for expression profiling of cell types in parallel, is labour- and resource-intensive. Single-cell RNA sequencing (scRNA-seq) is more robust to many of these factors, but is expensive, especially in a large-scale study of many subjects [5,6]. These bottlenecks mean many studies of immune cells use mixed cell populations, such as peripheral blood mononuclear cells (PBMC), which might hinder the discovery of genes that exert their roles in a cell-type-specific manner.

Computational deconvolution of cell specific transcriptomes from mixed cell RNA-seq data provides an alternative to address this challenge. It is generally hypothesised that expression at a given gene in a mixed cell sample is the summation of its cell-type-specific expression weighted by corresponding cell fractions [7,8].

$$m = H \times f \tag{1}$$

where $m$ is the vector of observed gene expression profiles of n genes, $H$ is a latent $n \times c$ matrix representing gene expression profiles in each of $c$ cell types and $f$ is a vector of cell fractions [7]. Note that while our goal here is to impute cell type specific expression profiles for each individual, $H$ is often treated as constant across a population, or some subgroup of the population (written $H1$). The sample-level cell-type expression profiles for a cohort must be indexed by individual, and may be written as $G$ ($n \times c \times k$) where $k$ is the number of samples. The initial aim of most deconvolution approaches is to estimate $f$, and many deconvolution methods have been developed to solve this equation [9–23], divided into supervised and unsupervised types depending on whether $H$ or $f$ is used to guide deconvolution [7].

Fraction deconvolution methods rely on pre-computed cell type reference/signature gene expression profiles $H$, and differ in regression models/optimising strategies employed to minimise the sum of the squares between fitted expression and $m$ [7], as well as their data pre-processing strategies and whether they allow for unknown cell types in the mixture. For example, both EPIC [12] and quanTIseq [15] employ constrained least square regression, while FARDEEP implements adaptive least trimmed squares to automatically detect and remove expression outliers that might lead to inaccurate f estimates [14] and CIBERSORT utilises linear nu-support vector regression that is robust to noise, unknown cell types in mixtures, and collinearity among closely related cell types in $H$ [10].

CIBERSORTx extends the functionalities of CIBERSORT to first estimate $f$ using a pre-defined subset of genes which are thought to distinguish cell types and for which predefined estimates of the subset of $H$ are supplied (again, assuming this profile is equal across individuals). In a second step, CIBERSORTx will then estimate the average cell-type specific gene expression in a set of samples, $H1$ ($n \times c$), conditional on $f$ using non-negative least squares (NNLS) regression [13]. Other supervised methods such as Rodeo [21] and csSAM [9] also estimate $H1$ assuming $f$ is known. If we write $M$ and $F$ for the matrix analogues of $m$ and $f$ for $k$ samples, unsupervised deconvolution methods directly decompose $M$ into $F$ and $H1$ simultaneously but require prior knowledge of the numbers of cell types and additional scRNA-seq expression data for annotating the results [16,18,19,22]. Differentially expressed genes can then be identified by estimating $H1$ separately by a condition of interest, e.g., disease status, with variance estimated via bootstrapping [6] or repeated $H1$ deconvolution with permutation processes [9], but covariates or quantitative outcomes can not be taken into account in this way. In this case, sample-level cell type gene expression is needed but additional constraints or assumptions are needed to estimate the sample-level expression given only $M$ and $F$ [13,23].

We identified five out of ten existing methods (S1 Table) that have developed strategies to impute sample-level cell-type gene expression: CIBERSORTx [13], CellR [22], MIND [17], bMIND [20] and swCAM [23]. CIBERSORTx assumes that each gene can be analysed independently and for each gene $i$ that is significantly expressed in at least one cell type, CIBERSORTx iteratively applies bootstrapped NNLS to estimate and refine cell-type expression coefficients for imputing $G_{i,\bullet,\bullet}$, based on solving the equation $G_{i,\bullet,\bullet} \times F = M_{i,\bullet}$, where $F$ is first estimated as above. CellR models sample-level cell-type gene expression as a function of RNA-seq read counts, assuming they follow a negative binomial distribution, and infers sample-level cell-type expression using a simulated annealing process [22]. MIND implements an expectation-maximization algorithm for sample-level cell-type gene expression by leveraging multiple transcriptomes from the same subjects [17]. bMIND [20], developed by the same authors of MIND, overcomes the limitation of multiple measurements per subject and uses a Bayesian mixed-effects model to estimate cell-type expression in each sample via Markov chain Monte Carlo sampling. swCAM [23] is built on debCAM [18], which does not require a signature matrix and estimates fraction and average cell-type gene expression in a convex analysis of mixtures framework. swCAM infers imputed cell-type gene expression for each sample using low-rank matrix factorisation, assuming cell-type expression variations across samples result from a small number of cell-type specific functional modules, such as transcription factor regulatory networks [23].

Prediction accuracy of cell fractions among deconvolution methods and factors affecting the performance have been extensively investigated based on synthetic data from scRNA-seq data sampled with designated cell proportions [24–26]. However, less is known about the accuracy of deconvolution-based approaches in imputing cell-type gene expression at the sample level. Here, we use RNA-seq data from mixed and sorted cell populations from the same individuals and pseudobulk datasets generated from eQTLgen scRNA-seq data [27] to examine the accuracy of CIBERSORTx, bMIND and swCAM. We excluded CellR, which relies on

predefined cell-type clusters in scRNA-seq data, and MIND, developed by the same authors of bMIND, which requires multiple bulk expression measurements per subject. We compare these domain-specific to general machine learning methods, multivariate LASSO and ridge, that are not dependent on the deconvolution equation. Instead, they can be used to estimate $G_{\bullet,j,\bullet}$, for each cell type $j$=1,...,$c$ independently given training data - samples in which both cell type specific and mixed cell transcriptomes have been measured. These machine learning approaches have not, to our knowledge, been used before in this context (Fig 1). Multivariate models differ from standard models by jointly modelling sets of genes, which we hoped would allow more accurate inference by exploiting correlation in expression between different genes.

## Results

The CLUSTER Consortium aims to use immune cell RNA-seq data to find transcriptional signatures which predict treatment response in childhood arthritis. In order to balance the competing goals of maximising both the number of patients studied and the number of cell specific assays from each patient within a fixed budget, we designed an RNA-sequencing experiment with PBMC samples included from all available subjects and specific immune cells from a subset of subjects, with the aim to use the subjects with coverage of both to learn rules to impute cell specific gene expression into the complete dataset. This design also allowed us to split our data into training (80 samples) and testing (between 52 and 71 samples depending on cell type) sets (Fig 2A) to compare the performance of potential imputation approaches.

### Estimation accuracy of sample-level cell frequencies

CIBERSORTx comes equipped with an inbuilt leukocyte gene signature matrix LM22, but also allows the creation of custom gene signatures. We created a custom signature using our sorted cell expression in the training set (Fig 2B). We deconvoluted CD4, CD8, CD14, and CD19 cell fractions from PBMC mixed cells based on different scenarios: CIBERSORTx with inbuilt (CIBX-inbuilt) and custom (CIBX-custom) matrices, bMIND using the custom matrix (bMIND-custom), debCAM using cell-type specific markers derived from expression in our sorted cell populations (debCAM-custom) and compared estimates of cell fractions to measures of ground truth derived from flow cytometry (Fig 2B).

Correlations were highest in CD14, followed by CD19, CD8, and CD4 regardless of methods and signature matrices (Fig 3). Generally speaking, CIBX-custom performed less well across all four cell types than the other three approaches, while CIBX-inbuilt and debCAM performed the best, although the exact ordering did vary between cell types. This difference between CIBX-custom and CIBX-inbuilt emphasises the importance of a well trained gene expression signature matrix. We note that CD14 predicted fractions were overestimated regardless of approach and CD4 generally under-estimated (Fig 3).

### Estimation accuracy of sample-level cell type gene expression

Our main goal was to compare accuracy in imputing sample-level cell-type gene expression from PBMCs for the four sorted cell populations: CD4 T cells, CD8 T cells, CD14 monocytes and CD19 B cells. In addition to predicted expression by CIBERSORTx using inbuilt and custom signature matrices, we derived cell-type expression profiles using true cell fractions (estimated by flow cytometry) with bMIND and swCAM algorithms (Fig 2B). Finally, we trained regularised multivariate LASSO and ridge models to predict cell-type expression using all genes. While CIBERSORTx predicts only a subset of the most confident genes (5779-11794 depending on cell type), all other methods were able to predict expression for all or almost all 18871 genes across cell types (S1 Fig).

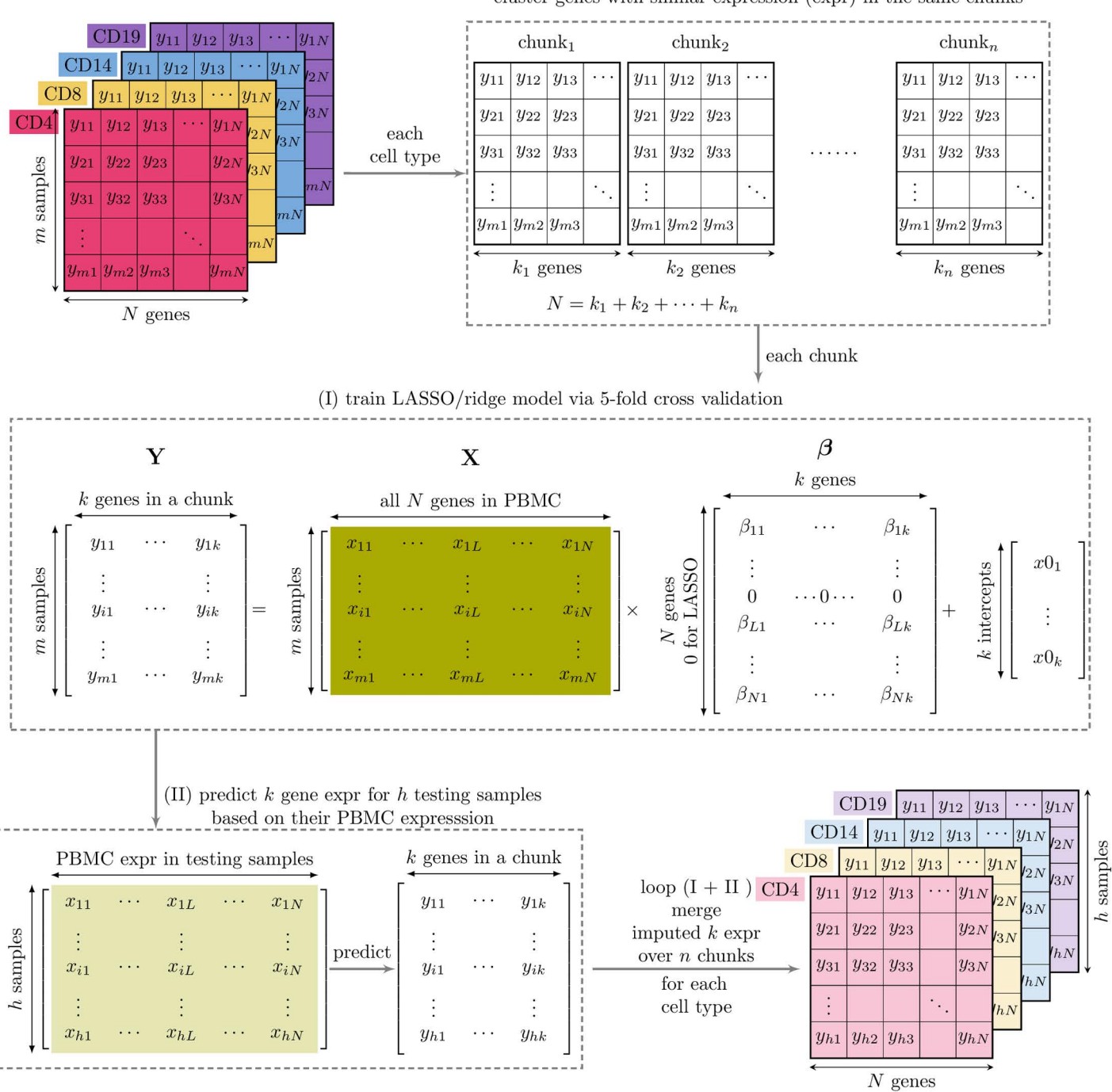

**Fig 1. Multi-response LASSO/ridge models for predicting sample-level cell-type expression.** We utilised gene expression data from pure cell types (such as CD4, CD8, CD14, and CD19) and a mixed cell type (such as PBMC), all obtained from the same subjects as our training data. For each cell type, we clustered genes with similar expression into chunks. For each chunk, we learned the expression associations between cell-type-specific target genes and predictor genes in PBMC using a multi-response LASSO/ridge model with five-fold cross-validation. The multi-response model includes a group penalty so that regression coefficients β for any given predictor may be shrunk to zero for all target genes. The non-zero β take different values corresponding to each target gene. This multi-response LASSO/ridge model was then employed to predict the expression of cell-type-specific target genes in testing samples based on their PBMC expression. The learning and prediction steps were repeated for all gene chunks for each cell type, and the predicted target gene expression was assembled on a per-cell type basis.

**A**

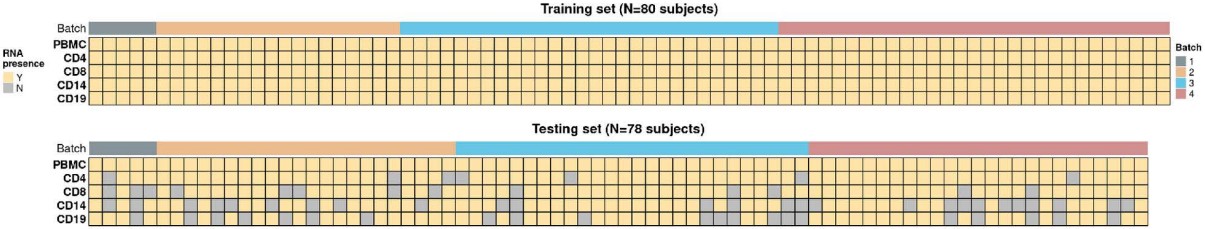

**B**

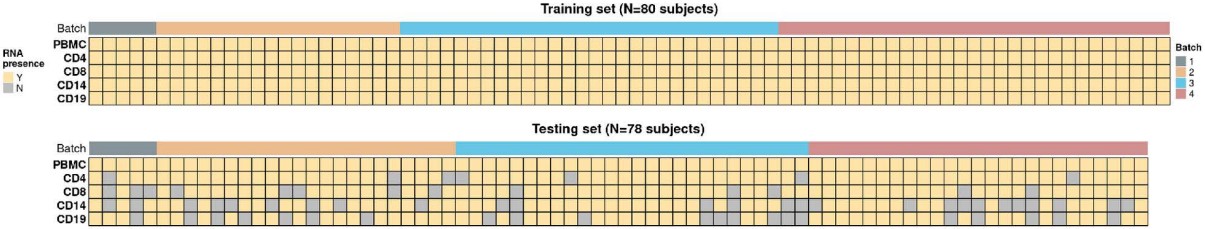

**Fig 2. Data and study design.** (A) CLUSTER samples by cell type (row) and subject (column). Cells are coloured based on the availability of RNA (Y for yes, N for no), and the top panel annotations indicate the RNA sequencing batch (Batch) (B) Data analysis workflow. Transcripts

per million (TPM) were calculated after excluding low-expressed genes. TPM from sorted cells (CD4, CD8, CD14, and CD19) from 80 training samples were used to generate custom signature genes using the CIBERSORTxFractions module. We deconvoluted the cell fractions from PBMC based on inbuilt and custom signatures using CIBERSORTx, using the custom signature genes with bMIND and cell-type specific genes using debCAM. Estimates of cell fractions were compared to the ground-truth cell fractions from flow cytometry, and we assessed fraction accuracy using Pearson correlation and RMSE (root mean square error). Next, we estimated sample-level cell-type gene expression based on inbuilt and custom signature matrices using the CIBERSORTx high resolution module. In parallel, we ran bMIND and swCAM, with the flow cytometry cell fractions, in a supervised mode for estimating cell-type expression. For each cell type, we trained a LASSO/ridge model on PBMC and sorted cells with 5-fold cross-validation and used this to predict cell-type gene expression in the test samples. We compared imputed cell-type expressions with the observed ones and evaluated and benchmarked the performance using Pearson correlation, RMSE and a novel measure, differential gene expression (DGE) recovery.

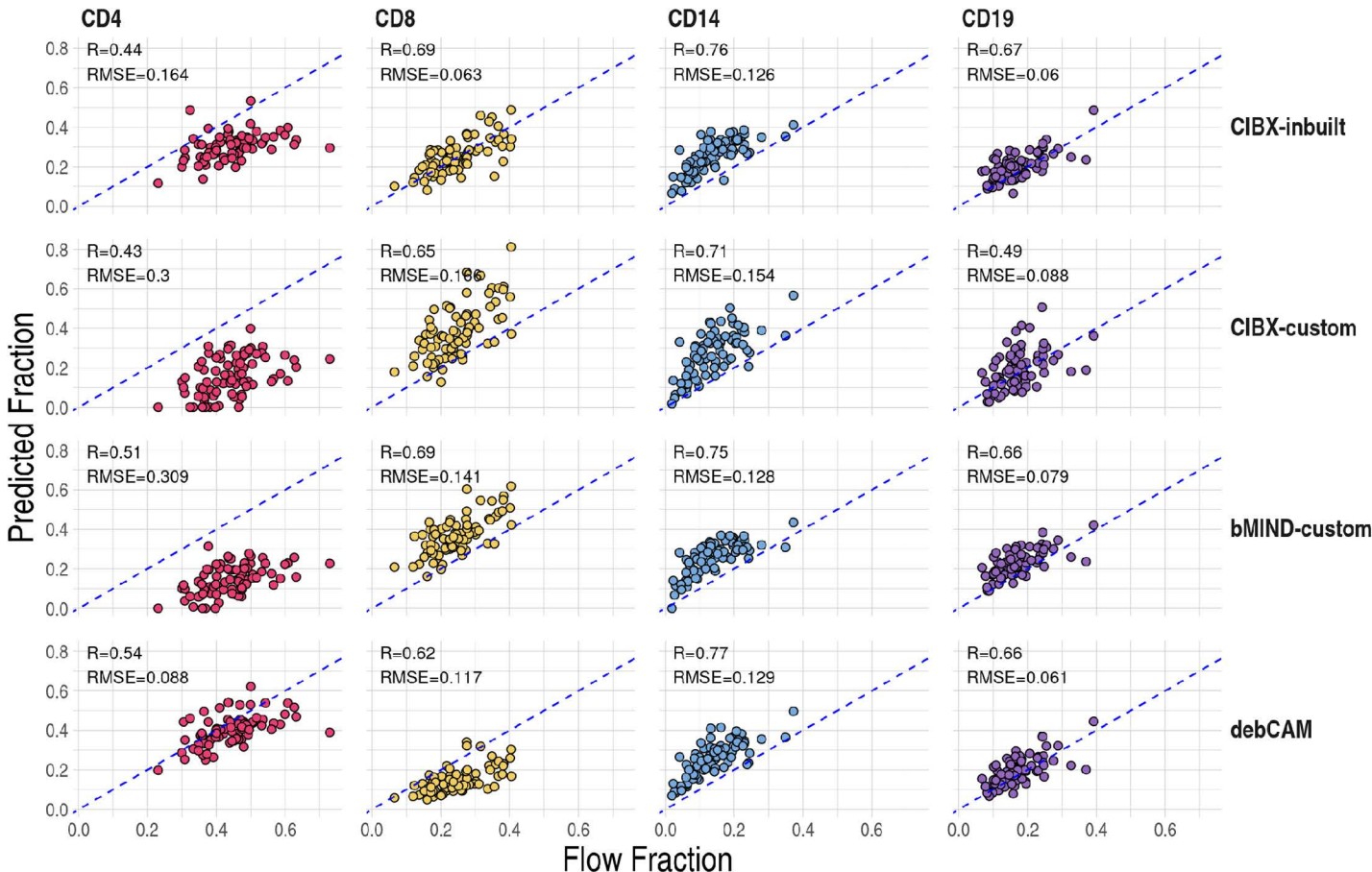

**Fig 3. Prediction accuracy of cell fractions by cell type (column) and approaches (row).** Pearson correlation (R) and root mean square errors (RMSE) were calculated between estimated fractions (y-axis) and flow cytometry measures (x-axis). Each point is a testing sample and dashed blue lines indicate y = x. CIBX-inbuilt: CIBERSORTx fraction deconvolution using the inbuilt signature matrix; CIBX-custom: CIBERSORTx fraction deconvolution using the custom signature matrix; bMIND-custom: bMIND fraction estimation using the custom signature matrix; debCAM-custom: debCAM fraction estimation using cell-type specific genes.

Despite the differing number of imputed genes among methods (S1 Fig), initial analysis comparing observed and predicted expression of genes in the same subjects suggested that all methods could predict cell-specific expression well, as judged by high correlation between observed and imputed expression (median r > 0.85) in the test data, although correlations were generally higher and root mean square error (RMSE) lower in LASSO and ridge than the other approaches (Fig 4A and 4B). To better interpret the correlations, we calculated

"baseline" correlations. These were estimated either between observed expression in one individual and estimated expression in the same cell type from a different individual, or between the observed expression across different subjects within the matched cell types. These were also high, and only marginally different from baseline correlations, limiting the utility of this measure to discriminate among methods (S2 Fig).

We therefore complemented this with a comparison of the observed and predicted expression across subjects for each gene. Correlation varied considerably between genes, irrespective of approaches (Fig 4C). All methods had comparable correlation per gene for each cell type, except for swCAM, which exhibited suboptimal performance for CD8, CD14, and CD19 (Fig 4C). Similar RMSE per gene was seen for each cell type across methods, although LASSO and ridge had slightly lower median values than other approaches (Fig 4D). Because different methods imputed values for different genes, we also compared methods restricting to the common gene sets predicted by all methods and found broadly the same ordering of methods by performance (S3 Fig).

## Recovery of significant genes in association testing

Despite correlation and RMSE being commonly used to assess predictive accuracy, they do not necessarily capture performance of predictions in intended downstream analyses. We therefore defined a new measure of performance, differential gene expression (DGE) recovery, which used simulated phenotypes that deliberately correlated with observed expression across a subset of genes and conducted DGE analysis in parallel using predicted and observed cell-type data. DGE recovery measured the degree to which significant and non-significant signals in the observed data could be correctly identified in the imputed data (S4 and S5 Figs). According to this measure, LASSO and ridge exhibited higher median values of the area under the receiver operating characteristic curve (AUCs) across cell types than the AUCs achieved by CIBERSORTx with inbuilt and custom, bMIND and swCAM. This pattern was consistent across four simulated scenarios: either dichotomous or continuous phenotypes with or without covariate adjustment (Fig 5). Different methods can predict expression for different numbers of genes (S1 Fig), and we confirmed the broad pattern was also consistent when restricting to the common genes predicted by all approaches (S6 Fig). More detailed examination in the dichotomous phenotype setting showed that, generally, LASSO and ridge had higher sensitivity than CIBERSORTx, bMIND and swCAM, but also lower specificity (S7A Fig). In all cases, imputed estimates of $\log_2$ fold changes were attenuated in the imputed data, with average slopes of 0.60-0.70 in CIBERSORTx inbuilt, 0.64-0.76 in CIBERSORTx custom, 0.66-0.85 in bMIND, 0.55-0.90 in swCAM, 0.69-0.76 in LASSO and 0.68-0.76 in ridge (S7C Fig).

To provide a broader range of scenarios to compare these methods, we constructed cell-specific bulk RNA-seq from single cell data from eQTLgen [27] and performed a similar analysis. We compared expression between cells stimulated with *C. albincas* for 3 hours to untreated cells, in three settings: either unadjusted for batch, adjusted using batch as a covariate, or adjusted using Combat-seq This allowed us to consider different models including covariates or different ways to account for batch effects. We varied training sample size from 20 (25%) to 80 (100%) samples. As the training sample size increased, AUCs generally increased for all approaches. LASSO and ridge regression consistently outperformed other methods across all cell types, regardless of whether raw or batch-corrected DGE results, in which batch effects were accounted for as a covariate or batch-corrected read counts by Combat-seq were used (Fig 6). This dominance persisted in further analysis of common gene sets (S8 Fig). All our results, taken together, suggest that regularised multivariate models performed better than the other three deconvolution-based methods.

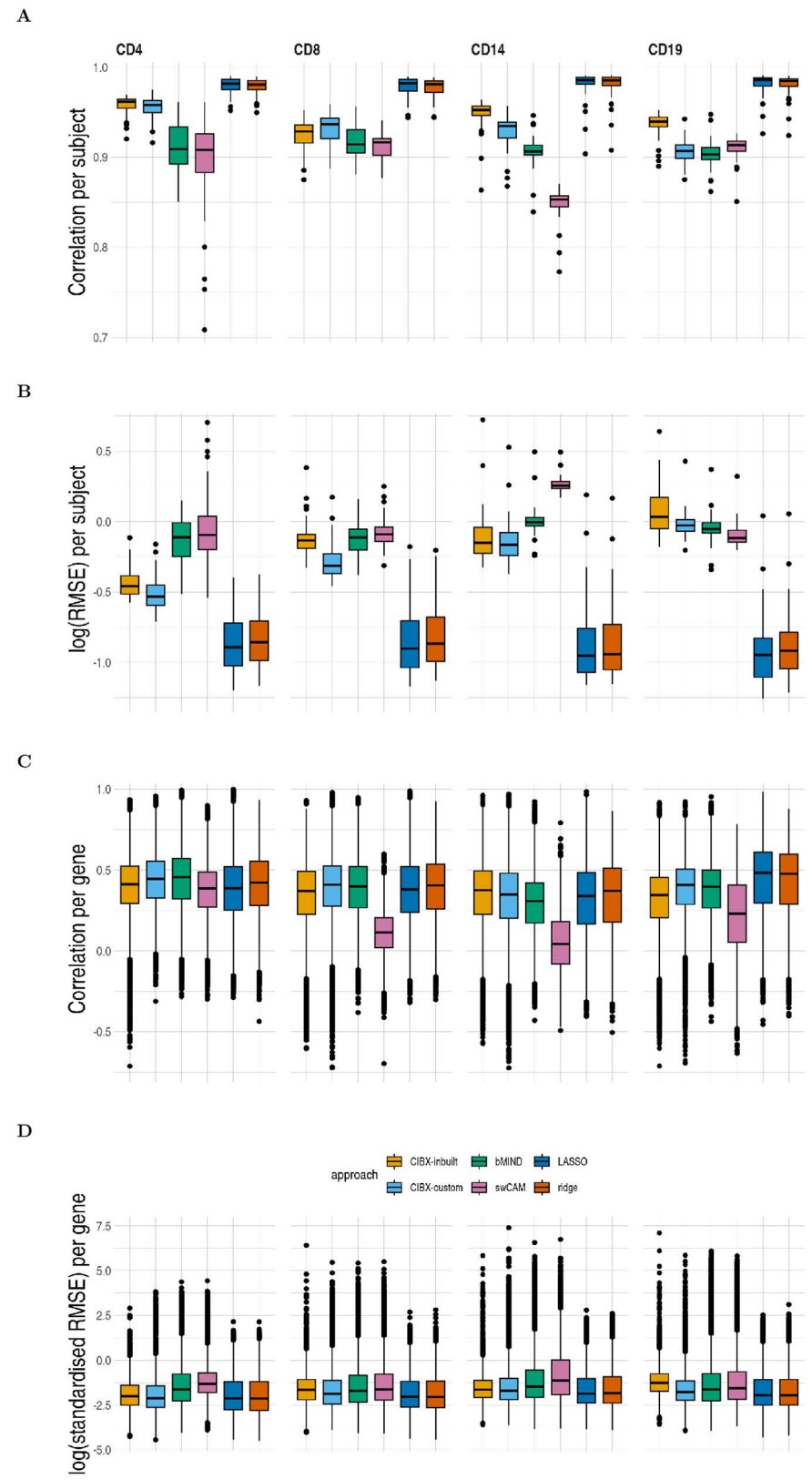

**Fig 4. Prediction accuracy of sample-level cell-type expression by approach.** (A) Pearson correlation and (B) log root mean square error (RMSE) comparing observed to predicted cell-type expression of genes from the same

subjects, one estimate per subject. (C) Pearson correlation and (D) log RMSE between observed and predicted cell-type expression across testing samples for each gene, one estimate per gene. RMSE was standardised by the average observed expression per gene. CIBX-inbuilt: CIBERSORTx expression deconvolution with the inbuilt signature matrix; CIBX-custom: CIBERSORTx expression deconvolution with a custom signature matrix derived from sorted cell-type expression in training samples; bMIND: bMIND expression deconvolution with flow fractions; swCAM: swCAM deconvolution with flow fractions; LASSO/ridge: expression predicted from regularised multi-response Gaussian models.

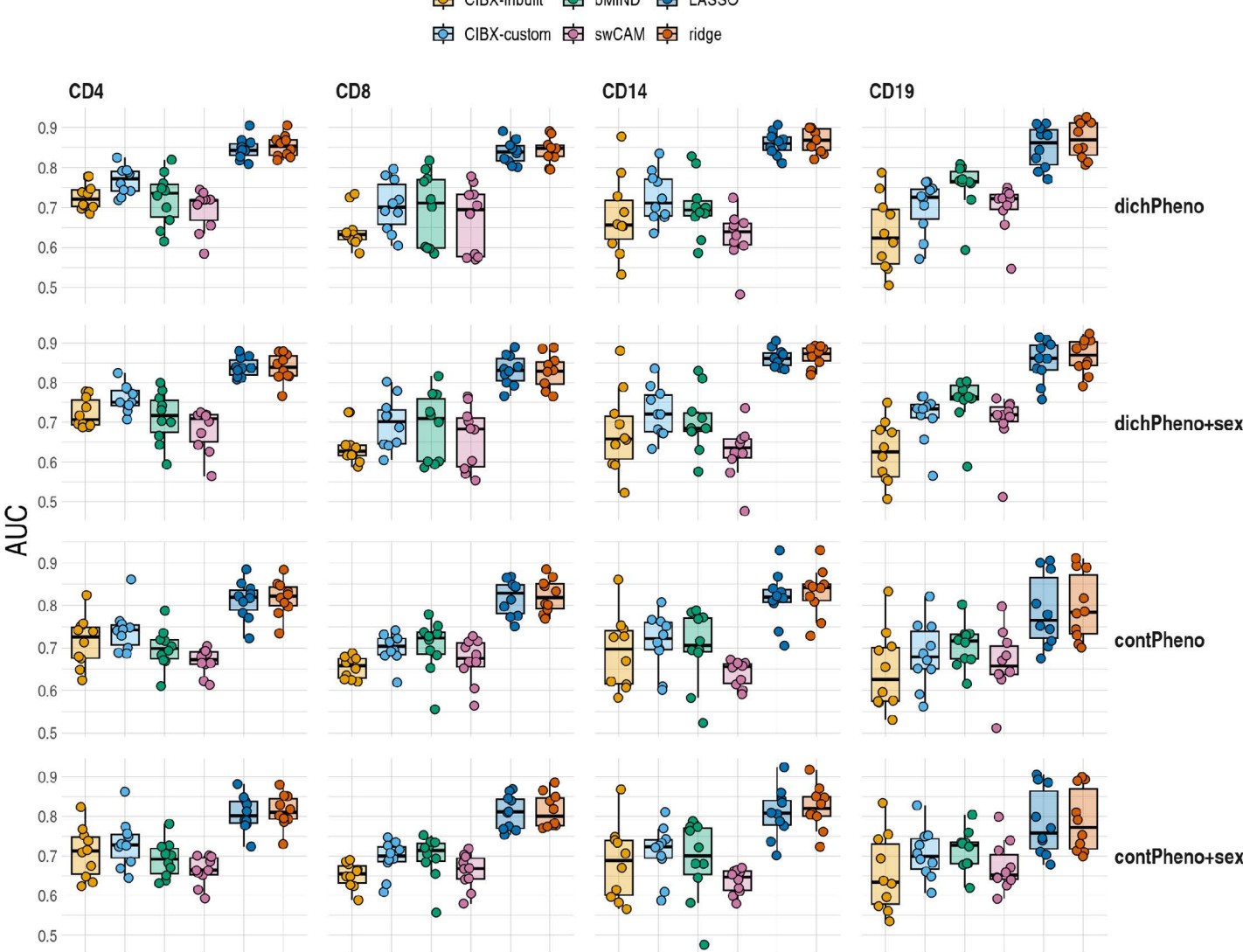

**Fig 5. Differential gene expression (DGE) recovery based on CLUSTER data.** Area under curve (AUC) distributions estimated in held out test data by approach and cell type (columns) for each scenario (rows). Scenarios differed in simulated dichotomous (dichPheno)/ continuous (contPheno) phenotypes, with/without sex as a covariate in the DGE models. dichPheno: dichotomous phenotype; dichPheno+sex: simulated dichotomous phenotype and sex; contPheno: continuous phenotype; contPheno+sex: continuous phenotype and sex. Each point is a simulated phenotype, and there are ten simulated phenotypes. For each simulated phenotype, the receiver operating characteristic curve and AUC were estimated by FDR fixed at 0.05 in the observed data and varied FDRs from 0 to 1 by 0.05 in the imputed data. Box plots showed the AUC distributions, with horizontal lines from the bottom to the top for 25%, 50% and 75% quantiles, respectively. CIBX-inbuilt: CIBERSORTx with the inbuilt signature matrix; CIBX-custom: CIBERSORTx with a custom signature matrix; bMIND: bMIND with flow fractions; swCAM: swCAM with flow fractions; LASSO/ridge: regularised multi-response Gaussian models.

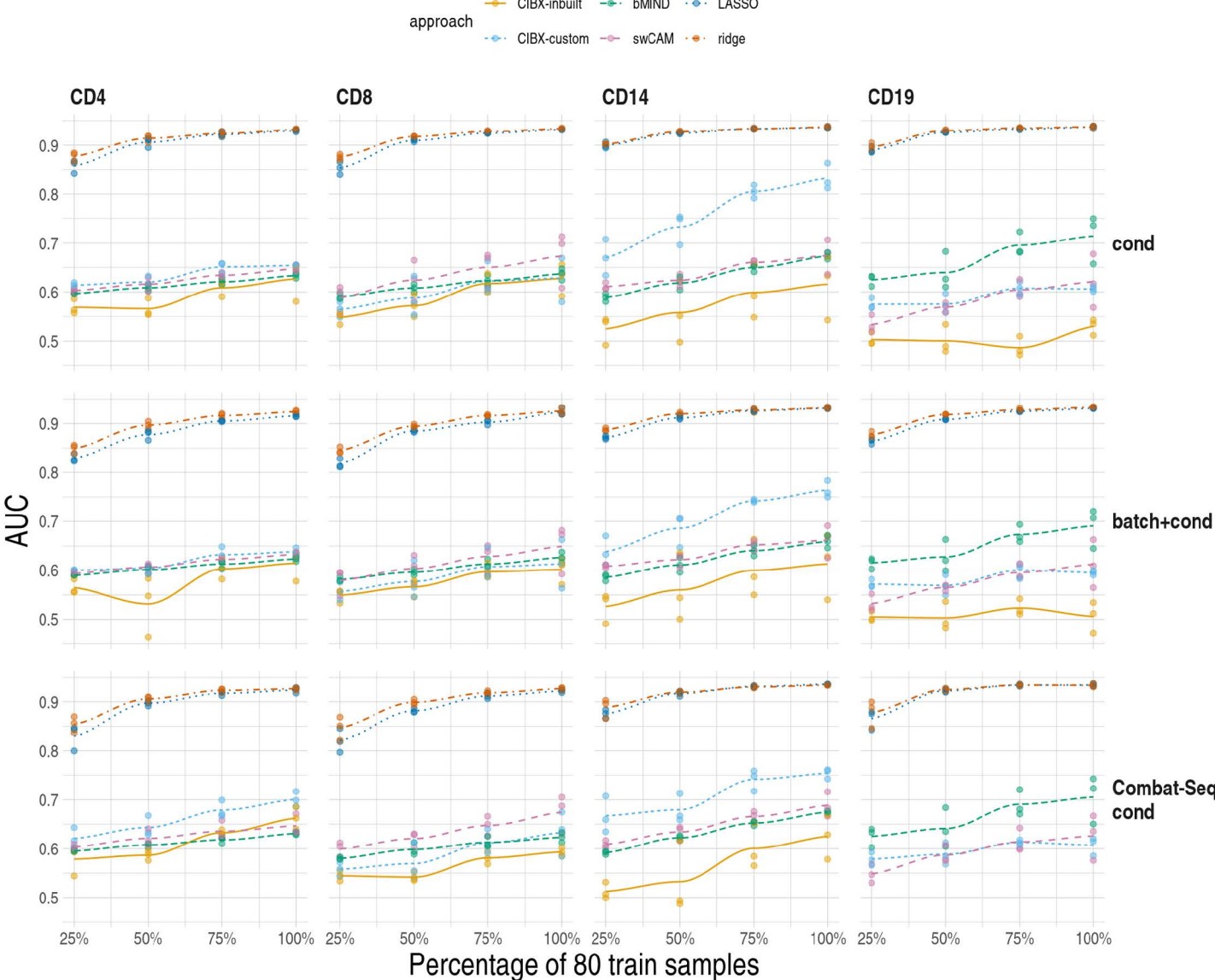

**Fig 6. Differential gene expression (DGE) recovery based on pseudobulk data.** We varied training sample size from 20 (25%) to 80 (100%) (x-axis) and quantified area under curve (AUC) in held out test data for DGE recovery (y-axis) by cell type (columns) for each scenario (rows). cond: raw aggregated read counts were used for DGE analysis of in vitro stimulation with C. albicans after 3 hours (3hCA) vs untreated (UT). batch+cond: same as cond, and with batch (V2 & V3 chemistry) as a covariate in the 3hCA vs UT DGE model. Combat-seq cond: Combat-seq batch-corrected read counts were used in DGE analysis of 3hCA vs UT. Each point is the result of one analysis, and three replicates were conducted for each training sample size. For each pseudobulk data, the receiver operating characteristic curve and AUC were estimated by FDR fixed at 0.05 in the observed DGE results and varied FDRs from 0 to 1 by 0.05 in the DGE results using imputed expression. Local polynomial regression fitting (loess) were plotted for each approach. Note that CIBX-inbuilt is not shown for CD19/Combat-Seq cond. This is because it was able to impute < 60 genes, compared to ~ 1,000 for CIBX-custom and > 11,000 for other methods, so estimates of AUC are very noisy. CIBX-inbuilt: CIBERSORTx with the inbuilt signature matrix; CIBX-custom: CIBERSORTx with a custom signature matrix based on pure cell expression in the training samples; bMIND: bMIND with true fractions; swCAM: swCAM with true fractions; LASSO/ridge: regularised multi-response Gaussian models.

## Computational performance

We compared CPU time and memory usage across methods in both datasets (S2 and S3 Tables). We found that typically CIBX was fastest, and bMIND required the least memory. swCAM and machine learning methods took longer and machine learning methods used

more memory. For all methods, CPU time increased approximately exponentially with respect to training sample size but memory usage remained the same (S9 Fig).

## Discussion

Using PBMC RNA-seq, sorted-cell RNA-seq, and flow cytometry data from the same individuals, our study investigated the accuracy of estimates of cell type fractions by the state-of-the-art domain-specific tools. All methods performed least well for CD4. Performance varied between cell types, suggesting that some cell type fractions (e.g., CD4) were consistently harder to estimate than others in this dataset. In addition, CIBERSORTx provided the most accurate estimates for CD8, despite not performing as well for CD4 compared to other methods. On the other hand, debCAM provided the best CD4 estimates but was less accurate for CD8. This suggests that accurately estimating fractions of these two related cell types remains challenging, possibly due to shared signature genes or a limited number of specific cell-type genes (S10A and S10C Fig). Both CIBX-inbuilt and debCAM generally outperformed bMIND and CIBX-custom; in particular the latter two produced several estimated cell fractions of exactly zero when observed data were clearly and substantially non-zero. We therefore recommend CIBX-inbuilt and debCAM for estimating cell fractions from mixed cell populations.

We provide a real data comparison of sample-level cell type specific expression imputation, including off-the-shelf machine learning methods, multivariate LASSO and ridge, as comparators. Correlation has been used to evaluate the accuracy of predicted cell-type expression, and good correlations per subject have been reported [13,20,23], consistent with our observations. However, we also found high correlations in between-subject comparisons (S2 Fig), which presumably reflects that cell type explains the greatest proportion of variability in gene expression. Good correlations at the sample level might not necessarily reflect accuracy at the gene level, as evidenced by low to moderate correlation per gene observed in our study, which was consistent with the findings of bMIND [20] and swCAM [23]. These suggest correlation is not an optimal measure of performance. In contrast, our proposed DGE recovery measure, which mimics DGE analysis and measures the capability to reconstruct DGE signals, could be more indicative than correlation. We observed better accuracy using LASSO and ridge than the three deconvolution-based approaches (CIBERSORTx, bMIND and swCAM).

To impute cell-type gene expression for samples, deconvolution methods first need an estimate of cell type fractions. CIBERSORTx estimates these from the RNA-seq data, and had good accuracy albeit with notable underestimation of the fraction of CD4+ T cells, while bMIND and swCAM utilised our flow cytometry-measured cell type fractions, which we expect to be more accurate representations of the sample composition used for RNA-seq. We might expect that using this additional data would allow bMIND and swCAM to be more accurate, but DGE recovery was comparable across deconvolution methods, although the number of genes with estimated expression does vary between methods (lower for CIBERSORTx). In contrast, LASSO and ridge, forms of penalised linear regression, use a one-step approach that does not rely on estimated cell fractions. Instead, it learns directly from a training set of PMBC and cell-type expression data. Rather than treat each gene as an independent problem, we used multivariate LASSO/ridge, batching genes with correlated expression in the target cell type to enable the solution for each gene in a batch to share information about which PMBC genes were important predictors.

Nonetheless, there are limitations in LASSO/ridge. Most obviously, LASSO/ridge requires a training dataset, consisting of bulk and cell-type gene expression data from the same subjects to train the model, unlike deconvolution-based methods that do not need such data. Moreover, LASSO/ridge demands high computational resources. Regarding CPU running time,

LASSO and ridge take up to 192.3 or 658.1 times as long as the fastest method CIBERSORTx. While bMIND is only 2.7 times slower than CIBERSORTx, swCAM is 385 times slower. Furthermore, LASSO and ridge require up to 3 or 11 times more memory usage than CIBERSORTx, while bMIND and swCAM only need 33% or 54% of CIBERSORTx's memory usage. We also note that predicted fold changes using imputed expression systematically shrunk the fold changes estimated from observed data across all methods, so that DGE analysis using imputed data can only be supported for detection of differentially expressed genes and direction of differential expression, not for unbiased estimation of fold changes.

LASSO and ridge are the machine learning methods we considered, and we have not attempted to optimise their performance. There is presumably potential to improve performance further, with consideration of how genes are batched for prediction, or by considering other approaches. Their better performance should motivate further exploration of non-domain specific methods in this space.

## Methods and materials

### Ethics statement

Data utilised in the study came from 158 subjects recruited in the CLUSTER consortium. Around 80% of subjects (N=126) have juvenile idiopathic arthritis (JIA); the rest are healthy controls from adults and children, and about 58% are female (N=91). Peripheral blood samples were obtained in accordance with the ethics approved by the London-Bloomsbury Research Ethics Committee (REC 05/Q0508/95, 95RU04, and 11/LO/0330). Formal written consent was obtained from the parent/guardian and the adult participants. The diagnosis for JIA followed the internationally agreed classification as described in [28].

Blood was collected in a heparinised tube and peripheral blood mononuclear cells (PBMC) were isolated by density gradient centrifugation with Lymphoprep (Stem Cell Technologies). The blood samples were collected from the JIA patients at different time points of treatment.

### Cell sorting

Isolated PBMC were sorted by cell sorter (BD FACSAria III, BD Biosciences) into different cell populations (S11 Fig) with CD4-BV711 (clone OKT4, Biolegend 317440), CD8-APC (clone SK1, Biolegend 344722), CD14- FITC (clone 61D3, eBioscience 11-0149-42), and CD19-PE-Cy7 (clone HIB19, Biolegend 302216). CD3-BV605 (clone OKT3, Biolegend 317322) was used to differentiate between T cell and non-T cell populations. Dead cells were excluded before sorting using 4,6-diamidino-2-phenylindole (DAPI; Sigma). Sorted cell purity was accessed and on average was>90%. For each subject, we divided CD4, CD8, CD14, and CD19 cell counts by the sum of CD4, CD8, CD14 and CD19 cells to obtain cell-type fractions.

### RNA sequencing and data processing

Unsorted PBMC and sorted immune cells were extracted with PicoPure RNA Isolation Kit (Applied Biosystems, KIT0204). The extracted RNA samples were sent to UCL Genomics for library preparation and sequencing.

RNA sequencing was carried out in four batches using Illumina NovaSeq6000. PBMC and sorted cell RNA-seq data were processed using the RSSnextflow (Resources), an RNA-seq pipeline customised for unique molecular identifiers (UMIs) tagged RNA-seq data built under the Nextflow framework [29]. Briefly, sequencing reads (2x100bp) were mapped to the reference genome GRCh38 using STAR aligner [30]. Two-passing mapping mode was used, with gene annotated features (Homo_sapiens.GRCh38.103.gtf) and the options of --twopassMode

Basic and --sjdbOverhang 99. Default parameters were used unless otherwise specified. Read PCR duplicates were identified based on alignment coordinates and up to 1 mismatched UMI sequence using the je suite tool [31]. After deduplication, aligned reads were summarised over the gene features using the featureCounts programme [32], and a read count table was generated for each batch.

We selected RNA samples that have RIN ≥ 5.0 and library concentration ≥ 4.5nM (~0.7ng/µL) for RNA sequencing. Illumina TruSeq mRNA stranded v2 and Roche Kapa mRNA Hyper-Prep were used to create libraries. Samples from the same subjects were sequenced in the same batch, and we employed Combat-seq [33] to minimise batch effects (S12 Fig). Read counts from across the four batches were analysed together. A total of 723 RNA-seq samples from 158 subjects with data on PBMC and at least one cell type were used in the downstream analysis. We filtered out genes with counts-per-million <0.836, equivalent to 10 read counts in our median library size of 11.95 million reads, in less than 96 samples. Also, genes were excluded if their total read counts across samples were less than 15. These filtering steps were conducted using the edgeR filterByExpr function [34,35], with cell type information as the group argument. After filtering out low expressed genes, transcripts per million (TPM), as recommended by the authors of CIBERSORT [36], was estimated and utilised as observed expression.

## Training/testing set

Of 158 subjects with PBMC gene expression data, 80 had complete RNA-seq data on all four sorted cells and PBMCs and formed the training set. The remaining 78 had data on PBMCs and partially complete gene expression data in sorted cells and were used as test samples, with numbers for each cell type of CD4: 71, CD8: 65, CD14: 52, CD19: 57 (Fig 2A). Sequencing batch did not differ significantly between training and testing sets (Chi-squared test p > 0.05, S13 Fig).

## eQTLgen pseudobulk data

Pseudobulk data were generated using scRNA-seq data downloaded from the eQTLgen consortium (https://eqtlgen.org/sc/datasets/1m-scbloodnl-dataset.html) as of 26 September 2024. This dataset includes PBMC samples from 120 healthy Europeans, sequenced under seven conditions: an unstimulated condition (UA) and three in vitro stimulation conditions with C. albicans (CA), M. tuberculosis (MTB), and P. aeruginosa (PA) after 3 hours and 24 hours. Two sequencing chemistries (V2 and V3) were employed [27].

We specifically utilised QC-ed read counts from V2 and V3, along with cell type information and conditions, to create the pseudobulk data. The number of genes differed between V2 and V3; therefore, 19927 genes common to both chemistries were retained. Additional genes were excluded due to having different chromosome annotations (N=35) or not being found (N=4762) in the gene annotation data (Homo_sapiens.GRCh38.103), resulting in 15165 genes remaining.

We utilised Jaakkola and Elo's method to generate pseudobulk data. Their approach first generates the cell-type fractions randomly based on a normal distribution, using the same mean and standard deviation (SD) observed in the original public dataset [21]. So, we estimated the relative fractions of CD4, CD8, CD14, and CD19 in the eQTLgen single cells for each condition and chemistry from cell barcode annotation files provided. We then averaged these fractions and calculated the corresponding SD across conditions and chemistries for each cell type. The average percentages (SD) were 42.74% (4.30) for CD4, 28.12% (2.40) for CD8, 26.29% (6.21) for CD14, and 2.85% (0.36) for CD19. We used these estimates to generate four cell fractions each sampled individual by generating random normal samples, and standardising these to sum to 1.

We simulated three datasets, each with a sample size of 160. Conditions (UA and 3hCA) and chemistry types (V2 and V3) were randomly assigned to the samples. For each sample, cell-type read count expression was summed over 5000 cells randomly selected from the corresponding cell-type pool at a given condition and chemistry. Bulk expression was calculated by summing the counts from CD4, CD8, CD14, and CD19 cells, all drawn from the cell-type expressions obtained in the previous step. The number of cells selected for each type was proportional to the fractions generated above.

We processed these three pseudobulk datasets using Combat-seq, incorporating the chemistry information as a batch factor. This approach mimicked a scenario where the batch effect was minimised during the data processing step. Both three original pseudobulk data and their Combat-seq counterparts were utilised to evaluate the performance of predicting sample-level cell type expression using different sample sizes of the training samples. For each of the six simulation datasets, we designated the first half of samples as the training set and the rest as the testing set. We then varied the sample sizes of training samples, from 25% (N=20) to 100% (N=80) by 25%, while keeping the same testing samples. All cells used to generate data for any sample were either from cells stimulated with *C. albicans* or untreated. We evaluated every trained model's ability to predict the outcome of in vitro stimulation in the held out test data, quantifying this by AUC. In this case, there were six pseudobulk datasets at each sample size: three from the original pseudobulk and three from Combat-seq processed datasets, resulting in a total of 24 pseudobulk datasets. For each pseudobulk data, low expressed genes were filtered out using the edgeR filterByExpr function on cell type information prior to TPM expression calculation. No. genes in these pseudobulk data ranged from 11964 to 12464.

## Signature gene matrices & cell-type specific genes

Two signature gene matrixes were used. CIBERSORTx in-built LM22 was derived from microarray gene expression in 22 purified leukocyte subsets [10]. We constructed a custom signature based on CD4, CD8, CD14, and CD19 TPM in the training subjects using the CIBERSORTxFractions module, with the default settings of "--G.min 300 --G.max 500 --q.value 0.01 --QN FALSE --single_cell FALSE" for sorted RNA-seq. For CLUSTER data there were fewer signature genes in the inbuilt matrix (N=547) than the custom one (N=1589), which we attributed to the dynamic ranges of gene expression measured by two different platforms, microarray for inbuilt and RNA-seq for custom (S10A and S10B Fig).Sorted cell expression in training samples was also used in debCAM to identify cell-type specific genes for fraction deconvolution. debCAM OVE.FC (one versus everyone - fold change) criteria of 1, 2, 5 and 10 were used (S10C Fig), and we selected 1247 cell-type specific genes from OVE. FC of 10 because this number was comparable to custom signature genes. Despite differences inbuilt/custom signature and debCAM cell-type genes separated testing samples well based on their cell types (S10D–S10F Fig).

For the eQTLgen data, a custom signature matrix was derived from synthesised expression in pure cells from the training samples using CIBERSORTx and was utilised later in CIBERSORTx for predicting sample-level cell-type expression.

## CIBERSORTx analysis

We ran CIBERSORTx locally with a token requested from the CIBERSORTx website (Resources). We ran the CIBERSORTxHiRes module for deconvolution, with RNA-seq default settings and "--variableonly TRUE" for only outputting genes with variation in expression across subjects. When the LM22 signature was applied, two additional arguments of "--classes" and "--rmbatchBmode" were specified to aggregate cell type expression for 11

major leukocytes based on the shared lineage of 22 leukocytes [10] and minimise measuring variations in gene expression introduced by platforms, respectively.

CIBERSORTx fractions of CD4, CD8, CD14, and CD19 for LM22 were derived from the sum of proportions in their shared-lineage leukocytes (S10A and S14 Figs):

- CD4 as the sum of the proportions of T cells CD4 naive, T cells CD4 memory resting, T cells CD4 memory activated, T cells follicular helper, and T cells regulatory (Tregs). T cells

- CD8 was used as CD8 fraction

- CD14 fraction as the sum of Monocytes, Macrophages M0, Macrophages M1, and Macro-phages M2

- CD19 as the sum of B cells naive and B cells memory

We then scaled the resultant proportions to the sum of 1. Imputed cell-type expression was log2 transformed for downstream evaluation.

## bMIND and debCAM/swCAM prediction

bMIND cell fraction deconvolution was carried out using the authors' bMIND function with our custom signature. Cell-type specific genes that were 10-fold over-expressed in one cell type compared to others, as previously described, were specified in debCAM AfromMarkers function for estimating cell fractions from PBMC mixture.

We followed the authors' instructions to run bMIND and swCAM in a supervised mode on true cell fractions, as measured by flow cytometry in the CLUSTER data or designated ones in eQTLgen data, in the same samples to predict cell type gene expression for samples. More specifically, bMIND predicted expression profiles were obtained from the bMIND function of the MIND package [20], which took log2(TPM+1) transformed PBMC expression and cell fractions as inputs.

swCAM consisted of two steps. In the fine-tuning step, we conducted 10-fold cross-validation, randomly removing one-tenth of gene expressions from the sample and gene expression matrix followed by imputing back expression missingness, to determine the optimal lambda with the minimum of RMSE between missing and imputed expression. The R script, script-swCAM-cv.R, was used. In the predicting step, lambda of 800, PBMC TPM expression, true cell fractions, and grouped cell expressions (cell types * gene matrix) derived from NNLS were used in the sCAMfastNonNeg function for imputing sample-wise cell type expression. All the R scripts and functions related to swCAM were obtained from the authors' GitHub repository https://github.com/Lululuella/swCAM [23]. We then log2 transformed the estimates of cell type expression for downstream analysis.

## LASSO/ridge training and prediction

Standard penalised regression approaches aim to predict a single variable given observations of multiple predictors. Ordinary least squares aims to identify $\beta$ that minimises

$$\sum_i^N (y_i - \beta_0 - \sum_j^p x_{ij}\beta_j)^2$$ for predictors $x_j$ and outcomes $y$ measured in subjects $i = 1,...,N$ . In our application, $y$ is the expression of a target gene in a specific cell type, $x_j$ is the expression of gene $j$ in PBMCs, $\beta_0$ is an intercept term and $\beta_j$ is the coefficient to predict $y$ from $x_j$ .

Penalised regression approaches minimise this quantity subject to constraints: $\sum_j |\beta_j| \leq \alpha$

(LASSO) or $\sum_j \beta_j^2 \leq \alpha$ (ridge). We could train a separate LASSO model using all genes in the

mixed cell data to predict each gene in a sorted cell in turn. However, given the correlations between expressions of different genes, we hypothesised that multi-response models may be more accurate given they allow sharing of information between the genes to be predicted. In a multi-response model, all genes in the mixed cell data are still used but now prediction is made for a set of genes. Information is shared as coefficients for any predictor are either zero for all target genes or non-zero for all target genes. However, note that non-zero coefficients have distinct values for each target gene. Thus multi-response models are appropriate to predict subsets of correlated genes (Fig 1).

We first used cell-type specific expression in training samples to define these subsets of correlated genes. We clustered genes into chunks on the basis of their expression profiles (in log2 TPM), with a size of up to 500 genes. Specifically, we performed a hierarchical clustering analysis on Euclidean distances between genes for each cell type in the training data. We grouped clustering dendrograms into the initial number of chunks, the minimum multiple of 500 to include all genes, using the R cutree function. For each chunk exceeding the size of 500 genes, we repeated clustering analysis on genes in a given chunk, followed by dendrogram grouping using the cutreeDynamic function [37] specified with "minClusterSize" of 250, if necessary, reducing by steps of 5, until all the resultant chunks met the desired size of < 500. The numbers of chunks (sizes) in the CLUSTER data were 74 (49-481) for CD4, 71 (71-493) for CD8, 76 (76-496) for CD14, and 83 (41-486) for CD19, respectively. In 24 pseudo-bulk datasets, the total numbers of gene chunks (respective size ranges) were 1358 (4-500) for CD4, 1306 (8-498) for CD8, 1533 (3-499) for CD14 and 1180 (2-497) chunks for CD19.

For each chunk, we fitted a penalised multi-response linear regression (LASSO/ridge) performed using PBMC expression in all genes to predict cell specific expression in the chunk genes, on the log2(TPM+1) scale. To ease the computational burden, the fine-tuned LASSO/ridge model from a 5-fold cross-validation based on the mean squared error (MSE) criterion was used to impute cell-type expression in the testing samples. Penalised modelling was carried out using the R glmnet package [38], for which "family=mgaussian", "type.measure =mse", and alpha=1 for LASSO/alpha=0 for ridge were specified in the cv.glmnet function for model training, and predict function was used for imputation.

## Performance measurement

Predicted cell fractions were compared to the flow cytometry estimates described above, and Pearson r correlations and root mean square errors (RMSE) were calculated for measuring performance. We calculated correlations and RMSE per gene between imputed and observed log2 transformed expression to evaluate and benchmark the predicted accuracy of expression among CIBERSORTx using custom and inbuilt, bMIND using custom, swCAM with cell-type specific markers, LASSO and ridge.

We simulated ten phenotypes that correlated with gene expression in the observed cell type data, including testing samples, using principal component analysis (PCA). First we conducted PCA of the fully observed gene expression data in the target cell type. From each of these we derived a continuous and a binary phenotype. The continuous phenotypes were defined as the PC value plus a standard normal error. The binary phenotypes were by dichotomising each PC with values >0 designated as 1 for pseudo-cases; otherwise, 0 for pseudo-controls. For each simulated phenotype, DGE analysis was carried out twice in parallel under the limma framework [39]:

1. in all observed data (test + training)

2. In observed data from the training samples + imputed data from the test samples which had observed data for comparison

We treated (1) as the ground truth, and (2) as the likely analysis adopted in any real world study. The expression matrix in (1) and (2) were used in the same limma DGE analysis. We utilised log2 transformed expression data and a predefined design matrix that incorporated simulated phenotypes (with and without sex for CLUSTER data) or treatment conditions (with and without batch for the simulation data) in the limma::lmFit function to fit the linear model, followed by empirical Bayes moderation of the standard errors using the limma::eBayes function. A false discovery rate (FDR) of 0.05 was set as the significance level for true signals in the observed data. We varied FDRs in the imputed data from 0 to 1 by steps of 0.05. Receiver operating characteristic analysis was carried out by combination of cell type, method and scenario using the pROC package [40].

All analyses were performed with R-4.3.3 [41] unless otherwise stated

## Resources

CLUSTER consortium-Childhood arthritis and its associated uveitis: stratification through endotypes and mechanism to deliver benefit, https://www.clusterconsortium.org.uk/

RSSnextflow workflow for processing RNA-seq FASTQ files to generate analysis-ready read counts https://gitlab.com/b8307038/rssnextflow

CIBERSORTx, https://cibersortx.stanford.edu/

bMIND, https://github.com/randel/MIND

swCAM, https://github.com/Lululuella/swCAM

## Code availability

Nextflow workflow, R scripts, and markdown files to run the analyses, to generate and to summarise results in this work presented here, https://gitlab.com/b8307038/sceqtlgen

## Supporting information

**S1 Fig. Overlap of predicted genes by CIBERSORTx using inbuilt (CIBX-inbuilt) and our custom signatures (CIBX-custom), bMIND, swCAM, LASSO and ridge.** Predicted genes were defined as those with variations in expression across subjects. For each panel (cell type), the right bar plot indicates the numbers of predicted genes (No.Pred.Genes) by approach, and the top bar plot demonstrates No.Pred.Genes common in different combinations of approaches (black dots), but not in the grey-dot approaches, if present. (PDF)

**S2 Fig. Distributions of Pearson correlations (y-axis) between observed and imputed expression across genes from the same/different (diff) subjects and correlation between observed expression in different individuals in matched cell types (ObsDiff) by cell type and approach.** One estimate per subject. CIBX-inbuilt: CIBERSORTx with the inbuilt signature matrix; CIBX-custom: CIBERSORTx with a custom signature matrix derived from sorted cell-type expression in training samples; bMIND: bMIND with flow fractions; swCAM: swCAM with flow fractions; LASSO/ridge: regularised multi-response Gaussian models. (PDF)

**S3 Fig. Prediction accuracy of sample-level cell-type expression by approach.** For each cell type, genes common across approaches were used. No. common genes are 3837 for CD4, 6274 for CD8, 5185 for CD14, and 2689 for CD19, respectively. (A) Pearson correlation and (B)

log root mean square error (RMSE) comparing observed to predicted cell-type expression of genes from the same subjects, one estimate per subject. (C) Pearson correlation and (D) log RMSE between observed and predicted cell-type expression across testing samples for each gene, estimate per gene. RMSE was standardised by the average observed expression per gene. CIBX-inbuilt: CIBERSORTx expression deconvolution with the inbuilt signature matrix; CIBX-custom: CIBERSORTx expression deconvolution with a custom signature matrix derived from sorted cell-type expression in training samples; bMIND: bMIND expression deconvolution with flow fractions; swCAM: swCAM deconvolution with flow fractions; LASSO/ridge: expression predicted from regularised multi-response Gaussian models. (PDF)

**S4 Fig. Comparisons of log 2 fold changes in genes between the observed (x-axis) and imputed (y-axis) data by method (column) and by recovery status (row).** DGE analysis was carried out using limma based on one of the simulated phenotypes. An FDR of 0.05 was used in both observed and imputed data here, and CD4 DGE recovery is shown. DGE results in each method (column) are the same, with coloured points for genes falling into that category of recovery status (row) and grey points for genes not belonging to the same category. Recovery status: No, Yes-NS, and Yes-Sig. Yes-Sig (sensitivity; Sen): differentially expressed genes in the observed data were also called significant in the imputed data, and the orientations of the effect sizes are the same in both data. Yes-NS (specificity; Spe): genes are called non-significant (NS) in both data. No (error; Err): misclassified genes; Err is calculated as the percentage of misclassified genes to the total number of predicted genes. (PDF)

**S5 Fig. Receiver operating characteristic (ROC) curves and estimated area under curve (AUC, numbers noted) by cell type (row) and approach (column) based on one simulated phenotype.** FDR was fixed at 0.05 in the observed data and varied from 0 to 1 by 0.05 in the imputed data. Dashed lines indicate y = x. (PDF)

**S6 Fig. Differential gene expression (DGE) recovery.** Area under curve (AUC) distributions by approach and cell type (columns) for each scenario (rows). Scenarios differed in simulated dichotomous (dichPheno)/ continuous (contPheno) phenotypes, with/without sex as a covariate in the DGE analysis of the common gene sets across approaches. No. common genes are 3837 for CD4, 6274 for CD8, 5185 for CD14, and 2689 for CD19. dichPheno: dichotomous phenotype; dichPheno+sex: simulated dichotomous phenotype and sex; contPheno: continuous phenotype; contPheno+sex: continuous phenotype and sex. Each point is a simulated phenotype, and there are ten simulated phenotypes. For each simulated phenotype, the receiver operating characteristic curve and AUC were estimated by FDR fixed at 0.05 in the observed data and varied FDRs from 0 to 1 by 0.05 in the imputed data. Box plots showed the AUC distributions, with horizontal lines from the bottom to the top for 25%, 50% and 75% quantiles, respectively. CIBX-inbuilt: CIBERSORTx with the inbuilt signature matrix; CIBX-custom: CIBERSORTx with a custom signature matrix; bMIND: bMIND with flow fractions; swCAM: swCAM with flow fractions; LASSO/ridge: regularised multi-response Gaussian models. (PDF)

**S7 Fig. Detailed examination of DGE recovery measures calling significance at FDR < 0.05 in both imputed and observed data.** (A) Distributions of sensitivity and specificity of DGE recovery by cell type and approach. Each point is a simulated phenotype. (B) R-squared (Rsq, y-axis) and (C) slopes of imputed log2 fold changes (FC) regression on observed effect sizes by approach. Each point is a simulated phenotype, coloured by cell type. CIBX-inbuilt: CIBERSORTx with the

inbuilt signature matrix; CIBX-custom: CIBERSORTx with a custom signature matrix derived; bMIND: bMIND with flow fractions; swCAM: swCAM with flow fractions; LASSO/ridge: regularised multi-response Gaussian models.
(PDF)

**S8 Fig. Differential gene expression (DGE) recovery based on pseudobulk data.** We varied training sample size from 20 (25%) to 80 (100%) (x-axis) and quantified area under curve (AUC) for DGE recovery (y-axis) by cell type (columns) for each scenario (rows). cond: raw aggregated read counts were used for DGE analysis of in vitro stimulation with C. albicans after 3 hours (3hCA) vs untreated (UT). batch+cond: same as cond, and with batch (V2 & V3 chemistry) as a covariate in the 3hCA vs UT DGE model. Combat-seq cond: Combat-seq batch-corrected read counts were used in DGE analysis of 3hCA vs UT. DGE analysis was performed in the training and testing samples and restricted to common gene sets predicted by all methods. Each point is the result of one analysis, and three replicates were conducted for each training sample size. For each pseudobulk data, the receiver operating characteristic curve and AUC were estimated by FDR fixed at 0.05 in the observed DGE results and varied FDRs from 0 to 1 by 0.05 in the DGE results using imputed expression. Local polynomial regression fitting (loess) were plotted for each approach. Noted that no data is shown for CD19/Combat-Seq cond because the common gene set predicted by all methods equals zero. CIBX-inbuilt: CIBERSORTx with the inbuilt signature matrix; CIBX-custom: CIBERSORTx with a custom signature matrix based on pure cell expression in the training samples; bMIND: bMIND with true fractions; swCAM: swCAM with true fractions; LASSO/ridge: regularised multi-response Gaussian models.
(PDF)

**S9 Fig. Resource metrics based on pseudobulk data.** (A) CPU time in minutes (B) memory GB usage in the nature logarithm scale (y-axis) across 25%, 50%, 75% and 100% percentage of 80 train samples (x-axis). For serial jobs, swCAM, LASSO and ridge, CPU time was summed together for the same pseudobulk data, and median of memory usage was taken per pseudobulk data. Each point is a pseudobulk data. Local polynomial regression fitting (loess) were plotted for each approach. CIBX-inbuilt: CIBERSORTx with the inbuilt signature matrix; CIBX-custom: CIBERSORTx with a custom signature matrix generated using pure cell expression of the training samples in the pseudobulk data; bMIND: bMIND with true fractions; swCAM: swCAM with true fractions; LASSO/ridge: regularised multi-response Gaussian models.
(PDF)

**S10 Fig. Differences between inbuilt and custom signature genes, and debCAM cell-type specific genes.** (A) Expression of inbuilt signature genes (N=547) in 22 leukocyte subsets, curated by the CIBERSORTx team from microarray gene expression. For each gene (row), expression is centred to the mean and scaled by the standard deviation across cell types (column). Columns are split by the collapsed classes. CD4: T cells CD4 naive, T cells CD4 memory resting, T cells CD4 memory activated, T cells follicular helper, and T cells regulatory (Tregs); CD8: T cells CD8; CD14: Monocytes, Macrophages M0, Macrophages M1, and Macrophages M2; CD19: B cells naive and B cells memory; PCs: Plasma cells; GammaT: T cells gamma delta; NK: NK cells resting and NK cells activated; Dendritic: Dendritic cells resting and Dendritic cells activated; Mast: Mast cells resting and Mast cells activated; Eos: EosinophilsPMN: Neutrophils (B) Expression of custom signature genes (N=1589), derived from our sorted-cell RNAseq expression in 80 training subjects using CIBERSORTx. Expression is centred and scaled across cell types (column) by gene (row). (C) Numbers (No.) of debCAM cell-type specific genes by cell type (row) and selection criteria (column). debCAM, which

does not have the signature matrix, selects the cell-type-specific genes, that are over-expressed in one cell type versus everyone (OVE). OVE fold change (FC) of 1, 2, 5 and 10 were used in our sorted cell expression of 80 training subjects. The first four principal components from PCA analysis of (D) inbuilt, (E) custom signature and (F) debCAM cell type specific gene expression in test samples. Each dot is a sample, coloured by cell type and shaped by sequencing batch.
(PDF)

**S11 Fig. Gating strategy for sorting cells into different immune cells.** Initial gating was performed with forward (FSC) and side (SSC) scatters to isolate lymphocytes (PBMC). Further gating with FSC-A and FSC-W was done to exclude doublets. Live cells were selected based on the gating with DAPI. The live cells were gated on CD3 to separate between CD3+ and CD3- cells. The CD3+ population was further gated for CD4 and CD8, whilst the CD3- population was gated for CD14 and CD19.
(PDF)

**S12 Fig. The first four principal components (PC) from PCA analysis of log2 (count-per-million) expression derived from (A) raw read counts (B) Combat-Seq batch-adjusted read counts in RNAseq samples used in this work.**
(PDF)

**S13 Fig. Frequencies (A) and percentages (B) of subjects by batch and training/testing set.**
(PDF)

**S14 Fig. Pearson correlations of cell fractions between 22 leucocyte (LM22) and ground-truth flow cell types.** Columns are LM22 cell subsets split by their merged classes (top annotation). Rows are ground-truth cell types. Each cell is coloured based on the strength of the correlation.
(PDF)

**S1 Table. Summary of existing deconvolution approaches.**
(PDF)

**S2 Table. Computational time and memory usage by approach based on the CLUSTER data.**
(PDF)

**S3 Table. Computational time and memory usage by approach based on the eQTLgen pseudobulk data.**
(PDF)

## Acknowledgments

We thank members of the Rheum Shared Seq (RSS) collaboration, George Robinson, Kathryn O'Brien, Elizabeth Ralph, Vicky Alexiou, Lucy Marshall, Emma Welsh, Meredyth G Ll Wilkinson, and Elizabeth C Rosser who contributed to sample processing, metadata, data generation, and curation. We thank the UCL Genomics facility for sequencing, specifically to Paola Niola and Tony Brooks, and UCL Flow Cytometry Facility for cell sorting, specifically to Ayad Eddaoudi, Machaela Palor, and Panagiota Constandinou. We also thank Dr George Hall and Prof Sergi Castellano for their support and advice. We thank all the patients, parents and families who allowed us to use data and donated samples, without whom this study would not have been possible. We extend our thanks to the research study coordinators, hospital staff, and extended research staff for their invaluable support in recruiting patients and working with families.

Some laboratory consumables were funded by research grants from Swedish Orphan Biovitrum (SOBI) Ltd and AbbVie Inc. This study acknowledges the use of Childhood Arthritis Response to Medication Study (CHARMS) the JIA Pathogenesis study (JIAP) both funded by Sparks UK, reference 08ICH09; and the Medical Research Council, reference MR/M004600/1; and the Adolescent Bioresource at the Centre for Adolescent Rheumatology at UCL UCLH and GOSH, part funded by the NIHR Biomedical Research Centre at GOSH.

**Members of the CLUSTER Consortium are as follows:**

Prof Lucy R. Wedderburn, Dr Melissa Kartawinata, Ms Zoe Wanstall, Ms Bethany R Jebson, Ms Freya Luling Feilding, Ms Alyssia McNeece, Ms Elizabeth Ralph, Ms Vasiliki Alexiou, Mr Fatjon Dekaj, Ms Aline Kimonyo, Ms Fatema Merali, Ms Emma Sumner, Ms Emily Robinson (UCL GOS Institute of Child Health, London); Prof Andrew Dick, (UCL Institute of Ophthalmology, London); Prof Michael W. Beresford, Dr Emil Carlsson, Dr Joanna Fairlie, Dr Jenna F. Gritzfeld (University of Liverpool), Ms Karen Rafferty, Ms Laura Whitty, Ms Jessica Fitzgerald; Prof Athimalaipet Ramanan, Ms Teresa Duerr (University Hospitals Bristol); Prof Michael Barnes, Ms Sandra Ng, (Queen Mary University, London); Prof Kimme Hyrich, Prof Stephen Eyre, Prof Soumya Raychaudhuri, Prof Andrew Morris, Dr Annie Yarwood, Dr Samantha Smith, Dr Stevie Shoop-Worrall, Ms Saskia Lawson-Tovey, Dr John Bowes, Dr Paul Martin, Dr Melissa Tordoff, Ms Jeronee Jennycloss, Mr Michael Stadler, Prof Wendy Thomson, Dr Damian Tarasek (University of Manchester); Dr Chris Wallace, Dr Wei-Yu Lin (University of Cambridge); Prof Nophar Geifman (University of Surrey); Dr Sarah Clarke (School of Population Health sciences and MRC Integrative Epidemiology Unit, University of Bristol); Dr Victoria J Burton, Dr Thierry Sornasse (AbbVie Inc.); Daniela Dastros-Pitei MD, PhD, Sumanta Mukherjee, PhD (GlaxoSmithKline Research and Development Limited.); Dr Michael McLean, Dr Anna Barkaway, Dr Victoria Basey (Pfizer); Dr Peyman Adjamian (Swedish Orphan Biovitrum AB (publ) (Sobi)); Helen Neale (UCB Biopharma SRL.); The CLUSTER Champions.

## Author contributions

**Conceptualization:** Wei-Yu Lin, Claire Deakin, Coziana Ciurtin, Lucy R. Wedderburn, Chris Wallace.

**Data curation:** Wei-Yu Lin, Melissa Kartawinata, Bethany R. Jebson, Hannah Peckham, Anna Radziszewska.

**Formal analysis:** Wei-Yu Lin.

**Funding acquisition:** Lucy R. Wedderburn, Chris Wallace.

**Investigation:** Wei-Yu Lin, Melissa Kartawinata, Bethany R. Jebson, Restuadi Restuadi, Hannah Peckham, Anna Radziszewska.

**Supervision:** Claire Deakin, Coziana Ciurtin, Lucy R. Wedderburn, Chris Wallace.

**Writing – original draft:** Wei-Yu Lin, Melissa Kartawinata, Chris Wallace.

**Writing – review & editing:** Wei-Yu Lin, Chris Wallace.

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
