## [Decision Letter · Decision Letter 0]

11 Jun 2024

Dear %TITLE% Wallace,

Thank you very much for submitting your manuscript "Penalised regression improves imputation of cell-type specific expression using RNA-seq data from mixed cell populations compared to domain-specific methods" for consideration at PLOS Computational Biology.

As with all papers reviewed by the journal, your manuscript was reviewed by members of the editorial board and by several independent reviewers. In light of the reviews (below this email), we would like to invite the resubmission of a significantly-revised version that takes into account the reviewers' comments.

As you see, all three reviewers found your work interesting and a good fit to PLOS CB, but they had important comments on the methods and on the writing. I am confident that taking these comments into account will significantly improve your manuscript.

We cannot make any decision about publication until we have seen the revised manuscript and your response to the reviewers' comments. Your revised manuscript is also likely to be sent to reviewers for further evaluation.

Sincerely,

Marc Robinson-Rechavi

Academic Editor

PLOS Computational Biology

Ilya Ioshikhes

Section Editor

PLOS Computational Biology

Reviewer's Responses to Questions

**Comments to the Authors:**

Reviewer #1: The authors investigate cell type deconvolution, where populations of cell types and/or their gene expression signatures are inferred from bulk RNA-seq samples. This investigation involves comparing multiple existing tools to the simple machine learning/statistical baselines of LASSO and Ridge regression. This topic is important and the paper is very well written. Overall, I think that the work is interesting and worth publishing but that the description could be expanded at some key points to ensure that the audience follows the precise steps taken and evaluations presented.

(1) Are the correlations for sample-level cell-type expression prediction being displayed over the same genes for each method or over different sets of genes? The authors indicate that CIBERSORTx focuses on only a subset of genes and other methods may also not make predictions for the same set of genes. It is challenging to compare methods if the performance is measured on different outputs.

(2) It was unclear to me why the "baseline" correlation was between the *observed* expression in one individual versus the *predicted* expression in another individual. It is true that this demonstrates some level of variance across individuals but, in my mind, the baseline here is compare the average gene expression profiles in training set individuals (for each cell type) versus the gene expression profiles in test set individuals (in matched cell types). Basically, if one did not use a machine learning model, how well could one do.

(3) Are the correlations for the sample-level cell-type expression prediction calculated on the raw counts or the log2 transformed data? The sections describing the methods indicate that the models are trained to predict log2 transformed data but the performance measurement section does not specify. Given the heteroskedastic nature of gene expression values, the more important comparison is likely on log2 transformed data.

(4) The chunking approach for LASSO/Ridge training is a bit confusing to me and it would be helpful if the paper could expand on what precisely is being done. The sentence beginning "Specifically, we performed a hierarchical clustering.." is a bit vague. Are the cell types in the training data or are the genes in the training data? Are expression values predicted for individual genes or just for these chunks and each gene is assumed to have the same expression? The main text says "Finally, we trained regularised... models to predict cell-type expression using all genes." That statement doesn't seem correct if you are only using these chunks. It is true that you use gene expression values from all genes to derive the chunks but the models themselves only see these chunks, is that correct? Can the cross-validation of models be explained in more detail with respect to what the exact training, validation, and test sets are?

Reviewer #2: Wei-Yu Lin et al present a comparative assessment of methods to impute cell-type specific gene expression profiles from bulk RNA-seq samples, including state of the art methods and two non-domain specific regression models (LASSO and RIDGE). They provide a summary of the state of the art methods and different metrics to evaluate the performance.

I believe that the comparative assessment is relevant to the field, and the community would benefit from an unbiased analysis of existing methods. However, in order to be considered for publication I believe the manuscript needs to be improved under different points of view, that I highlighted in this review.

Abstract and author summary:

I think that the author summary is much more clear than the abstract. I would personally suggest to rewrite the abstract with the same information flow followed in the author summary.

In fact, starting the first sentence in the abstract mentioning DGE might be misleading, since it suggests that the paper addresses the problem of inferring DGE.

Instead, in the author summary the addressed problem is explained and motivated, and the impact on DGE inference is subsequently addressed, and this makes it easier for the reader to grasp the concepts presented in the article.

Introduction:

The introduction needs to be re-written with a more clear and schematic flow.

- Authors make a distinction between methods that work with f and m vectors and those that infer M and F as matrices for multiple samples. If it is necessary for understanding the addressed task, authors should start by describing the two settings more clearly, and they should clarify which tools work with multi-sample matrices and which ones infer f and m vectors for one sample.

- Authors should also clarify which methods keep f or H fixed, and which ones infer both of them simultaneously.

- In my opinion the description of methods such as Rodeo and csSAM, which are not involved in the comparative assessment, should be removed.

- Finally I would make a figure with a summary of the methods, indicating what values are inferred and what approach is employed by each of the presented tools.

Mathematical notation: I think that the mathematical notation needs to be more thorough.

- Why on page 5 H became referred to as H1?

- I may have not understood some key element in the paper, but I am finding it difficult to understand from the manuscript what matrix is inferred by CIBERSORTx: if the signature matrix H is provided as input, what are the expression values inferred by the tools? I think that this could be clarified with a more thorough mathematical notation to describe each task performed by the methods in the comparison.

For example, authors could employ a similar description as that presented in CIBERSORTx methods, where it is explicitly reported that the imputed gene expression matrix is derived by first inferring the abundance vector F using a signature matrix B containing a subset of markers, and then solving the system of linear equations M = H x F.

- Overall, I think it is necessary to use a specific mathematical notation for each concept discussed in the manuscript: if H is the imputed gene expression matrix, which is different from the signature matrix (denoted with B in CIBERSORTx manuscript, methods, page 11 of the PDF), this signature matrix needs to be explicitly described.

- The two regression methods (LASSO and RIDGE) analysed need to be more systematically described. Given that they have not been presented in the context of gene expression imputation before, authors need to provide a solid mathematical representation of the regression models.

- A graphical notation about the matrices could be added in the same figure summarising the compared methods.

- Authors should write explicitly in the methods the vectors and formulas used to calculate the performances, keeping the notation consistent with that adopted in the description (M = H x F)

- The simulations of phenotypes for the DGE analysis needs to be explained more clearly.

Comparative assessment results:

- Regarding the LASSO and RIDGE regression, in order to state that they provide better estimates of imputed gene expression values I think it is fundamental to test them on external datasets that are not related to those employed in the CLUSTER project.

- For a more extensive comparison of the tools, authors should include tests on synthetic data. One possibility is to use a single-cell RNA sequencing simulation tool such as Splatter, which can estimate simulations parameters from real data and can thus provide simulated cells consistent with real datasets.

Single cell counts can be aggregated to produce a pseudo-bulk expression vector per sample, which can be used to test the different methods discussed in the manuscript.

- Authors should also provide a better analysis of the computational time and memory usage of the methods, varying the number of subpopulations and the number of samples.

Reviewer #3: Lin et al. present an interesting manuscript focused on cell-type specific expression imputation and show that one-step ML models perform better for this task than the otherwise typically applied deconvolution-based approaches on their dataset. I think this is a relevant paper and fits the scope of PLOS Comp. Bio in general. I have some concerns about reproducibility and computational things, which should be easily remediable.

Otherwise, the dataset the authors use is extremely interesting and important to the research community, but I am not including it in my review here, as it is, from what I understand, not claimed as a contribution of this particular paper.

My other major concern(s) center around downstream tasks of cell-type specific expression imputation. First, I think the authors should either present some downstream results of their proposed methods (i.e., Ridge/Lasso for cell-type specific expression imputation) on non-simulated data to highlight the biological relevance of their contribution or lean more heavily into the methodological side, which for me would mean additional datasets/simulations as well as baselines. In either case, I would appreciate further comments on relevant downstream tasks of cell-type specific expression imputation - while of course, the raw imputation is already very interesting methodologically, the authors focus on DGE as a downstream task, in which case I do not understand off the bat why their methods cannot also be compared to cell-type specific DGE methods and benchmarks [1, 2, 3]. There is mention of quantitative traits and/or covariates, hinting, I think, that some of the methods (such as [2] or csSAM) are restricted to a two-group comparison setting. I don't see how the DGE recovery downstream task actually encapsulates this, however? If it accounts for covariates, it would be useful to emphasize this and perhaps still make a limited comparison, either by restricting all methods to the two-group setting (since I understand all simulated phenotypes are binary) and/or ignoring other covariates.

Otherwise I found the paper rigorous in that no overly grandiose claims were made, and weaknesses of the proposed methods (in particular strong reliance of the ML methods on [large-ish] training datasets which may not always be available) were acknowledged.

I summarize my comments overall below as follows:

Major:

- I like that the authors have paid attention to computational considerations, but have some concerns regarding these. (i) To what extent were different methods parallelized? I see that e.g., for glmnet, a parallel backend was registered, whereas e.g., for bMIND I see no mention of parallelization. Please clarify this - IMO, if some of the methods cannot be parallelized (adequately), all methods should be compared unparallelized for fairness. (ii) Why are all numbers integers in Supplementary Table? E.g., the RAM for Ridge and Lasso looks unbelievably high if it's really in GB. (iii) Ideally, the authors should rerun the methods multiple times to evaluate their computational features and indicate mean +- standard deviation or similar. (iv) Why is CiberSortX inbuilt missing from Supplementary Table 1?

- I like the premise of this study but feel that to round it out, it would ideally need (i) further biological downstream analysis, using the proposed methods and/or (ii) more detailed experiments, turning it into an almost quasi-benchmark. Following on (i), the paper mentions "The CLUSTER Consortium aims to use immune cell RNA-seq data to find transcriptional signatures which predict treatment response in childhood arthritis.". I think the paper could be much stronger if the authors could give e.g., some examples they can discover downstream in their data by using their proposed approach with Ridge or Lasso that would not have been discoverable with the other compared to deconvolution approaches.

- Also, following up on (ii) of my previous point, it would (alternatively or in addition to the downstream biological investigation of the dataset via the newly proposed methods) be interesting to evaluate alternative baselines and/or datasets. Concretely, I wonder a bit about the main downstream tasks of imputation of cell-type specific expression. As I see it, the main downstream task (also as presented in the paper) is differential expression. In that case, however, the authors should also compare to dedicated methods for cell-type specific differential expression. I am not an expert on this, but there seem to exist several benchmarks [1, 2] and methods [3] (see my comment above re limitations to the two-group comparison setting). I suggest the authors either clarify/expand their choice of downstream tasks and/or add methods for cell-type specific differential expression to the analysis, ideally also in other datasets (e.g., [1, 2] both have simulated datasets that could be used).

Minor:

- I appreciate that the authors have made code (and processed data) available. That said, I think some things are missing from the code repo to ensure optimal reproducibility. First, it would be important to note the versions of all R packages (as well as R [I realize this is mentioned in the paper, but IMO the repo should be self-contained concerning reproduction]). If this is integrated into Nextflow (e.g., via conda/renv) even better, but this is not necessary. However, at a minimum, there should be a text file listing all package versions (effectively just having the sessionInfo separately as opposed to in the HTML where it may not be immediately visible, I mean), ideally even a renv.lock file or comparable, which makes it feasible for interested researchers to reproduce the author's results in the same environment. Second, the repo needs additional documentation as to how the results can be reproduced. In particular, it should cover these three questions: (i) Which commands should be executed to set up everything (e.g., download data, install requirements if any, ...) (ii) Which commands should be executed to rerun the experiments (iii) Which commands should be executed to reproduce the figures (from what I can tell this is already in the RMDs, but would be still worth pointing out explicitly since I didn't see this or the session info until I looked at the HTML). [In case I missed any of these, perhaps just highlight them more clearly in the repro as I guess I am somewhat close to your "average target audience" wrt reproduction]

- I am quite unclear about how the "simulation" for DGE recovery was performed, I suggest the authors re-check this section to make it more clear. (i) I suggest thinking about using a different word than simulation, since, unless I misunderstand, the authors are assigning samples to phenotypes, but using the same expression. This confused me, as a simulation to me implies that counts are being simulated as well. (ii) What was the reasoning behind PC_i > 0 for stratification? Such that one gets phenotypes with continually weaker signals that are balanced in sample size between the two phenotypes? (iii) Does it make sense to include the training samples in the ground truth? This shouldn't change the methods ranking since it applies to all methods equally, but I would prefer the authors perform the data only on test samples to prevent leakage. (iv) "The expression matrix used in (1) and (2) were the same in the limma DGE analysis." Can the authors clarify this sentence? Doesn't 2) use imputed data for at least some samples while 1) should only have observed data? (v) To clarify: Was the DGE performed per cell type, or not? My understanding was that it had been, but then the ROC sentence sounds like only the ROC curves were calculated per cell type.

- The terminology used for RNA-seq is slightly inconsistent at times (RNA-seq vs RNAseq).

- I find it fairly unfamiliar to see Ridge written in all capitals. For the Lasso, this is common as it is an acronym, but I have rarely seen it for Ridge.

- I am curious why the authors chose to correct for batch-effects. I realize that the Combat-Seq paper claims that its outputs are compatible with the count nature of common DGE methods, but I think it's still not obvious that this works (significantly) better than including batch as a covariate (especially if there is comparably little batch effect). In particular, Supplementary Figure 7 before correction looks quite fine to me even pre-batch correction.

- In some figures, CiberSortX is just indicated by "inbuilt" and "custom". Please add CiberSortX to this somehow as "inbuilt" and "custom" are not immediately obvious to be CiberSortX at first glance (I realize this is in the caption, but it makes scanning the figure significantly more confusing the first time reading it).

- I understand that the train/test split was stratified by batch (see e.g., Supplementary Figure 1). Have the authors also looked at the distributions of JIA and gender in the train/test and perhaps also stratified their results by either of these to make sure that no effect goes overlooked due to these? Or were these included as covariates in the DGE (in which case this was not clear/I didn't see this in the text explicitly).

- Can the authors comment on the fact that they only performed one split (i.e., no cross-validation)? I think it's fine since the datasets are very large for this type of data, but it could still be a good sanity check (as a Supplementary Figure at least) to make sure that the results are not too dependent on the specific split chosen.

- It would be useful to flesh out the methods section for "LASSO/RIDGE training and prediction". In particular, it may not be clear to all readers that the only way that a multitask Lasso/Ridge differs from independent (per task, i.e., per gene here) Ridge/Lassos is via the (grouped) regularization term (e.g., forcing all tasks to have the same non-zero coefficients for the Lasso), so it would be useful to write down the objective function to clarify this, I think.

- Please also give additional details on which DGE workflow in limma was exactly carried out.

**Have the authors made all data and (if applicable) computational code underlying the findings in their manuscript fully available?**

Reviewer #1: Yes

Reviewer #2: Yes

Reviewer #3: Yes

PLOS authors have the option to publish the peer review history of their article (what does this mean? ). If published, this will include your full peer review and any attached files.

**Do you want your identity to be public for this peer review?** For information about this choice, including consent withdrawal, please see our Privacy Policy .

Reviewer #1: No

Reviewer #2: No

Reviewer #3: No
---

## [Decision Letter · Decision Letter 1]

16 Dec 2024

PCOMPBIOL-D-24-00571R1

Penalised regression improves imputation of cell-type specific expression using RNA-seq data from mixed cell populations compared to domain-specific methods

PLOS Computational Biology

Dear Dr. Wallace,

Thank you for submitting your manuscript to PLOS Computational Biology. After careful consideration, we feel that it has merit but does not fully meet PLOS Computational Biology's publication criteria as it currently stands. Therefore, we invite you to submit a revised version of the manuscript that addresses the points raised during the review process.

I especially agree with the need to correct for small mistakes of writing in the newly added text. You may treat all other points of reviewer 2 as optional, although I ask you to consider them carefully and provide a detailed responses.

Please submit your revised manuscript within 30 days Feb 15 2025 11:59PM. If you will need more time than this to complete your revisions, please reply to this message or contact the journal office at ploscompbiol@plos.org. Please include the following items when submitting your revised manuscript:

We look forward to receiving your revised manuscript.

Kind regards,

Marc Robinson-Rechavi

Academic Editor

PLOS Computational Biology

Ilya Ioshikhes

Section Editor

PLOS Computational Biology

**Journal Requirements:**

Please ensure that the funders and grant numbers match between the Financial Disclosure field and the Funding Information tab in your submission form. Note that the funders must be provided in the same order in both places as well.

**Reviewers' comments:**

Reviewer's Responses to Questions

**Comments to the Authors:**

Reviewer #1: The authors have addressed all my concerns.

Reviewer #2: I appreciate the revisited version of the manuscript, especially the effort to make the mathematical notation more clear has made the paper easier to follow.

However, I still think that a few points need to be addressed further.

1. The definition of the LASSO and RIDGE regression needs to be further clarified. In fact, while I the equation with the predictor and observed variable is enough to summarise the regression, since the authors report in the manuscript that these general ML models have never been applied in this context and they are independent on the deconvolution equation, I believe it is necessary to explain what the predictor variables and their coefficients beta correspond to.

I might be missing a point but to me this is not clear. For example, are all x-s in the regression binary variable values corresponding to each gene, while the betas correspond to their expression?

2. Given that the procedure to apply LASSO/ridge regression involves different steps (e.g., the hierarchical clustering of genes), I think that a graphical representation of the method would increase the clarity of the manuscript.

3. Regarding the new test on eQTL data, what data is used to compute the AUROC measure? More specifically, when 75% of the samples are used for training, is the remainder 25% used for testing? Or is the AUROC computed on the same 75% points used for training?

Minor comments:

- There are few typos throughout the text. For example on page 21

Information is shared a coefficients for any predictor are either zero for all target genes or non-zero for all target genes

- On page 6: "We and compare the these domain-specific to general machine learning methods"

**Have the authors made all data and (if applicable) computational code underlying the findings in their manuscript fully available?**

Reviewer #1: Yes

Reviewer #2: Yes

PLOS authors have the option to publish the peer review history of their article (what does this mean? ). If published, this will include your full peer review and any attached files.

**Do you want your identity to be public for this peer review?** For information about this choice, including consent withdrawal, please see our Privacy Policy .

Reviewer #1: No

Reviewer #2: No

**Figure resubmission:**
---

## [Editor Report · Decision Letter 2]

7 Feb 2025

Dear %TITLE% Wallace,

We are pleased to inform you that your manuscript 'Penalised regression improves imputation of cell-type specific expression using RNA-seq data from mixed cell populations compared to domain-specific methods' has been provisionally accepted for publication in PLOS Computational Biology.

Best regards,

Marc Robinson-Rechavi

Academic Editor

PLOS Computational Biology

Ilya Ioshikhes

Section Editor

PLOS Computational Biology

---

## [Editor Report · Acceptance letter]

PCOMPBIOL-D-24-00571R2

Penalised regression improves imputation of cell-type specific expression using RNA-seq data from mixed cell populations compared to domain-specific methods

Dear Dr Wallace,

I am pleased to inform you that your manuscript has been formally accepted for publication in PLOS Computational Biology. Your manuscript is now with our production department and you will be notified of the publication date in due course.

With kind regards,

Zsofia Freund
